# CVCC Model: Learning-Based Computer Vision Color Constancy with RiR-DSN Architecture

**DOI:** 10.3390/s23115341

**Published:** 2023-06-05

**Authors:** Ho-Hyoung Choi

**Affiliations:** School of Dentistry, Advanced Dental Device Development Institute, Kyungpook National University, Jung-gu, Daegu 41940, Republic of Korea; chhman2000@msn.com; Tel.: +82-010-6771-6680

**Keywords:** computer vision color constancy, scene illuminant color, illumination estimation, RiR-DSN architecture

## Abstract

To achieve computer vision color constancy (CVCC), it is vital but challenging to estimate scene illumination from a digital image, which distorts the true color of an object. Estimating illumination as accurately as possible is fundamental to improving the quality of the image processing pipeline. CVCC has a long history of research and has significantly advanced, but it has yet to overcome some limitations such as algorithm failure or accuracy decreasing under unusual circumstances. To cope with some of the bottlenecks, this article presents a novel CVCC approach that introduces a residual-in-residual dense selective kernel network (RiR-DSN). As its name implies, it has a residual network in a residual network (RiR) and the RiR houses a dense selective kernel network (DSN). A DSN is composed of selective kernel convolutional blocks (SKCBs). The SKCBs, or neurons herein, are interconnected in a feed-forward fashion. Every neuron receives input from all its preceding neurons and feeds the feature maps into all its subsequent neurons, which is how information flows in the proposed architecture. In addition, the architecture has incorporated a dynamic selection mechanism into each neuron to ensure that the neuron can modulate filter kernel sizes depending on varying intensities of stimuli. In a nutshell, the proposed RiR-DSN architecture features neurons called SKCBs and a residual block in a residual block, which brings several benefits such as alleviation of the vanishing gradients, enhancement of feature propagation, promotion of the reuse of features, modulation of receptive filter sizes depending on varying intensities of stimuli, and a dramatic drop in the number of parameters. Experimental results highlight that the RiR-DSN architecture performs well above its state-of-the-art counterparts, as well as proving to be camera- and illuminant-invariant.

## 1. Introduction

The human visual system (HVS) is incredibly versatile in adjusting to changes in illumination sources [1,2]. The HVS has the ability to automatically modulate the receptive field size of visual neurons and achieve color constancy effortlessly. By simulating the adaption of HVS to changes in illumination sources, CVCC research has been focused on constructing highly reliable CVCC architectures and has also made remarkable progress in scene understanding and human and object detection, to name a few examples [3].

For a pixel of three colors—red, green, and blue—the intensity value ρ(x,y) at an (x,y) location is a function of three factors: illuminant distribution I(x,y,λ), surface reflectance R(x,y,λ), and camera sensitivity S(λ). The dependent claim is described as follows:(1)$$\rho \left(x,y\right)=\int_{\lambda }^{}I\left(x,y,\lambda \right)R\left(x,y,\lambda \right)S\left(\lambda \right)d\lambda$$
where λ refers to the wavelength. The CVCC approach aims at projecting the uniform global illuminant I(x,y,λ) at the sensor spectral sensitivity S(λ), represented as follows:(2)$$I=I\left(x,y\right)=\int_{\lambda }^{}I(x,y,\lambda )S(\lambda )d\lambda$$
where I is assumed to be a constant across the scene.

The CVCC approach is a two-stage process: estimating the illuminant color as in Equation (2) and restoring the image by multiplying the estimated illuminant color and the biased color image.

In an attempt to improve the accuracy and reliability of CVCC systems, this article presents a novel CVCC approach that introduces the RiR-SDN architecture. As its name implies, it has a residual network in a residual network (RiR) and the RiR houses a dense selective kernel network (DSN). A DSN is composed of selective kernel convolutional blocks (SKCBs). The SKCBs, or neurons herein, are interconnected in a feed-forward fashion. Every neuron receives input from all its preceding neurons and feeds the feature maps into all its subsequent neurons, which is how information flows in the proposed architecture. In neuroscience, it is a generally accepted fact that visual cortical neurons in human vision modulate the receptive field size to adapt to changes in illumination sources and achieve color constancy. However, this predominant trait of the HVS received little attention in the CVCC community, even though the primary focus of CVCC research has been on imitating how the HVS perceives constant colors of objects regardless of changes in illuminant conditions. Inspired by the remarkable trait of the HVS, the proposed RiR-DSN architecture has designed SKCBs to be able to modulate the filter kernel size, imitating the automatic modulation of the receptive field size in human vision, depending on varying intensities of stimuli. In the RiR-DSN, the *l*th SKCB receives l input that is the total feature maps of all preceding SKCBs and feeds the feature maps into all its subsequent SKCBs, or the L−l number of SKCBs. It is a general observation in the CVCC space that an increasing number of neurons and connections leads to higher performance. Motivated by the observation, the DSN of the proposed architecture is designed to have a residual block in a residual block, increasing its depth and complexity.

The key contributions of this article may be summarized as follows: (1) The proposed CVCC approach takes a step forward in the CVCC space by creating a deeper and more complex CVCC architecture and demonstrating that the proposed architecture outperforms its conventional counterparts with significant improvements in both estimation accuracy and computational efficiency, as evidenced by a large number of experiments with reliable standard datasets. The architecture also proves to be camera- and illuminant-invariant. (2) The proposed RiR-DSN architecture is a novel, unique attempt to mimic color constancy in the HVS. To the best of our knowledge, it is the first unique approach in that the proposed RiR-DSN architecture adopts SKCBs, which are designed to modulate the filter kernel size, imitating the automatic modulation of the receptive field size in human vision depending on varying intensities of stimuli. Section 2 discusses the previous works in detail, and Section 3 elaborates on the proposed RiR-DSN architecture for achieving color constancy. Section 4 evaluates the quantitative and qualitative results of estimating the illumination with the proposed architecture. Finally, Section 5 presents the conclusions of this paper. 

## 2. Previous Works

CVCC research has a history of several decades and has been primarily interested in estimating and removing color cast from digital images. The CVCC approaches fall into five categories: statistics-based, gamut-based, physics-based, learning-based, and biologically inspired approaches. In the CVCC realm, the statistics-based approach has the longest history and is the most widespread methodology. All statistical methods carry with them one or more assumptions and are required to validate their assumptions. Likewise, it may fairly be said that there are as many assumptions as there are statistics-based CVCC proposals. The underlying assumption among them is the Retinex theory, aka the Gray-World assumption, which hypothesizes that the mean of all pixel values of a well-corrected image is achromatic [4,5]. A large number of statistics-based CVCC proposals follow the Retinex theory, for instance, the GW and GE approaches. The GW approach assumes that the color of a well-corrected image averages out to gray. However, the assumption has a problem [4]: the performance is highly affected by color changes, and the corrected image suffers from a color bias caused by a larger uniform color region. While most statistics-based approaches are tried and developed based on the Retinex theory, the Max-RGB approach sets a different hypothesis that every image contains at least one white region, reflecting the chromaticity of the light source, and accordingly predicts that the maximum red, green, and blue colors of an image are the chromaticity of the scene illuminant [5]. Inevitably, however, the Max-RGB approach does not perform up to the Retinex theory, and it has the disadvantage of requiring massive data. At the same time, the Max-RGB approach has not overcome the color bias issue of previous statistics-based research. It also suffers from color bias caused by the dominant color of an image. As suggested in ref. [6], the Max-RGB approach takes a step forward by adopting the subsampling process. This framework uses random pixel selection and improves the Max-RGB approach. The SoG approach has been developed on the assumption that the Minkowski norm-p of a scene is achromatic [7]. Minkowski norm-p represents the mean or power average of the image data. The algorithm performs best with a Minkowski norm-p of 6. As suggested in ref. [8], the GE (Gray edge) approach uses the edge information to perform CVCC. The approach is based on the hypothesis that the mean derivative of the color intensity values is achromatic. This technique is the product of distilling insights from prior, well-known studies such as SoG (Shades of Gray), Max-RGB, or GW, and uses the first- and second-order derivatives to reach the white balancing of a given image. The approach applies an LPF (low-pass filter) and successfully removes noise, but computational efficiency suffers consequently. As proposed in ref. [9], weighted GE is designed to perform color rendering, factoring in the edge information of all objects in a scene.

Next, the gamut-mapping approach is first proposed by Forsyth [10], which presumes that only a few colors are observable under a source illuminant, i.e., the canonical gamut. It means that color changes are coming from variations in the color intensity values of the source illuminant. The technique is intended to achieve CVCC for images taken under unknown illumination and the predetermined canonical gamut. The approach delivers superior performance to its statistics-based counterparts in many cases. However, it comes with higher computational costs. Finlayson develops an extended, 2D version of the gamut mapping-based approach to reduce implementation complexity in chromatic space [11]. Further, Finlayson et al. [12] suggest a 3D approach. The 3D approach shows incremental progress as compared to its 2D counterpart. As in ref. [13], an effective implementation framework is proposed with the use of convex programming. The framework equals the performance of the first gamut-mapping approach. Mosny et al. [14] introduce a gamut mapping approach using a simple cube instead of a convex hull to increase estimation accuracy. Gijsenij and his colleague [15] propose a diagonal offset model to analyze the gamut-mapping approach [16]. The authors also suggest multiple gamut-mapping approaches based on various n-jets. In the gamut mapping community, the best performance is demonstrated by what Gijsenij and Gebers proposed [17]. The authors use Weibull parameters such as grain size and contrast to detect image features, and demonstrate that their approach performs 20% better than its best-performing counterparts.

The physics-based approach uses the dichromatic reflection model to estimate illumination based on physics-based interactions between a light source and an object. With the assumption that all RGB pixels of any surface correspond to dots in a plane of color space based on the RGB color model, this approach deals with the challenging task of extracting the reflection of a surface and has a color-clipping problem [18,19,20]. Finlayson and his colleague [21] try to model possible illuminations by using the dichromatic reflection model and the black-body locus to reflect the surface pixels. 

In CVCC research, the learning-based approaches employ as many machine learning models as possible to predict the illuminant color cast [22,23,24,25,26]. Baron [22] proposes a new CVCC approach by performing CVCC in log-chromaticity. The author adopts localization methodology, uses the scaling of color channels in a given image, and enables transformation into a log-chromaticity histogram. This is the first attempt to apply the convolutional neural network-based approach to achieving CVCC. Bianco and his colleagues [23] introduce an illumination estimation approach using a CNN. Their CNN is composed of a single neuron, a fully connected network, and three output nodes. The approach performs contrast stretching using the image histogram and then generates and integrates the activation values to predict the illuminant color cast. Another CNN-based approach is introduced as in ref. [24]. This approach adopts Minkowski pooling to develop a deep neural network. The fully connected neural network generates highly trustworthy feature maps and contributes to achieving white color balancing as a result. Drew and his colleague [25] propose a CVCC approach with a log-relative chromaticity planar, named “Zeta image.” It is worth noting that this algorithm does not require a lot of learning data and tunable parameters unnecessarily. The proposed method surpasses its unsupervised counterparts. Joze and Drew [26] present an exemplar-based approach that uses the neighboring surface to predict local source light, and they use learning data with weak RGB intensity values and texture features. Xu and his colleagues [27] also present a global-based video improvement approach that increases the visual experience of multiple regions of interest (RoIs) within an image or video frame in a meritorious manner. Their approach creates a global tone-mapping (TM) curve for the entire image by using the features of its diverse RoIs. As a result, various regions of the processed images are enhanced to a reasonable and meaningful level. Chen and his colleagues [28] introduce an intra- and inter-constraint-based (A + ECB) video quality improvement approach that investigates the RoI with an AdaBoost-based algorithm. This approach computes the average and standard deviation of the resulting RoI to explore the features of different RoIs, and then makes a global piecewise TM curve for the entire frame. They demonstrate meaningful and significant enhancement of the visual quality of diverse RoIs within the image.

Finally, there are several biologically motivated CVCC models that use machine learning in computer vision to imitate the core abilities of the human visual system (HVS) [29,30,31]. Gao and his colleague [29] develop an HVS-based color rendering approach and show that their approach has made incremental progress. Zhang and his colleagues [30] introduce an approach for achieving CVCC that is inspired by the HVS. The proposed approach reproduces the color processing in the retina of the human brain by trying to remove the influence of illuminant color cast rather than accurately estimating the illumination color cast. The approach outperforms its state-of-the-art counterparts. Akbrania et al. [31] propose a CVCC approach with the use of two overlapping asymmetric Gaussian kernels. Their proposed network is able to scale up and down the kernel size depending on contrast changes due to neighboring pixels. The authors have taken the closest approach to how the visual neuron changes the receptive field size. 

As discussed above, extensive CVCC approaches have been studied and proposed in order to estimate illuminant color cast and achieve color constancy in computer vision. They have made significant progress and performance improvements by optimizing conditions and assumptions. However, in most cases, they have a data dependency problem, which raises an effectiveness concern, and the resulting images are biased toward a dominant, color-constant region. To mitigate some of the limitations, this article presents a novel CVCC approach using the RiR-DSN architecture.

## 3. The Proposed Method

To achieve computer vision color constancy (CVCC), it is vital but challenging to estimate scene illumination from a digital image, which distorts the true color of an object. Estimating illumination as accurately as possible is fundamental to improving the quality of the image processing pipeline. Hence, the proposed CVCC approach is designed to achieve the maximum possible accuracy in inferring the illuminant color from a given image. This section elaborates on what the proposed RiR-DSN architecture is and how it works. 

By reproducing the ability of HVS to perceive colors consistently despite changes in illumination sources, CVCC is one of the underlying processes in the contemporary image processing pipeline for imaging sensors or devices [32]. There are two parts to the process. The first part is to estimate the illuminant color. Here, a single or multiple-illuminant vector is inferred from a given captured image. The estimated illuminant vector is a three-element value representing red, green, and blue color channels from a target image. The second part is to render the target image by multiplying the estimated illuminant color by the biased color image, which is called chromatic adaptive transformation. Ultimately, the rendered image is expected to look as if it was captured under standard D65 illumination. 

To this end, the proposed RiR-DSN architecture is built as a deeper and more complex architecture than the original residual block to improve estimation accuracy. Figure 1 shows a typical (a) basic architecture and its (b) basic block [33]. The basic block can come in various forms, depending on researchers and their architecture. With an image $x_{0}$ as input, the CNN architecture consists of L neurons, each of which carries out a non-linear transformation $H_{l}(\cdot )$, where l refers to an index of the neuron. A conventional feed-forward CNN architecture links the output of *l*th neuron as an input to the (*l* + 1)th neuron [34], formulating the following neuron transition: $x_{l}=H_{l}(x_{l-1})$. The ResNet features skip-connection, which performs non-linear transformations with the use of an identify function: (3)$$x_{l}=H_{l}\left(x_{l-1}\right)+x_{l-1}$$

Figure 2 shows the proposed RiR-DSN architecture designed to improve overall estimation accuracy. In the DSN of the proposed RiR-DSN, the $l^{th}$ selective kernel convolutional block (SKCB) has l input, which is the feature map of all its preceding SKCBs. A SKCB is made up of three phases: Split, Fuse, and Select. The Split phase generates a multi-path for layers that use diverse filter kernel sizes. Next, at the Fuse phase, selection weights are given to the feature map information, and the weighted feature map information moves to the next layer. Finally, at the Select phase, all the feature map information merges [35]. These feature maps are delivered to all subsequent SKCBs, or the L−l number of SKCBs. This is different from a basic CNN architecture in which a layer receives an input from its preceding layer, meaning the input is the output of the preceding layer and the output of the layer is fed into the next layer as input. However, what they have in common is that both SKCB and a basic CNN architecture receive $x_{0},\ldots ,x_{l-1}$ as an input. It is worth noting that an SKCB is able to scale up and down the filter kernel size depending on varying intensities of stimuli. The filter kernel size in the SKCB can be likened to the receptive field size in the HVS, which is scaled up and down to adapt to different illumination sources. Let the concatenation, $\left[x_{0},\ldots ,x_{l-1}\right]$, of feature-maps generate 0,…,l−1 neurons, and the dependent claim is described as follows [36]:(4)$$x_{l}=H_{l}\left(\left[x_{0},\ldots ,x_{l-1}\right],\right)$$

Additionally, the DSN combines a residual block in a residual block (RiRB), as shown in Figure 2, which is motivated by general observation that an increasing number of neurons and their connections lead to improving performance in CVCC research. The proposed CVCC approach uses the RiR-DSN architecture as the basic block in Figure 1. The proposed RiR-DSN architecture has potential applicability for many application fields.

## 4. Experimental Results and Evaluations

This section validates the proposed CVCC approach with RiR-DSN architecture by experimenting with several public and standard datasets and proves that it is illuminant- and camera-invariant.

Experiments are conducted using benchmark datasets: the Gehler and Shi dataset [37] of 568 images of a substantial variety of indoor and outdoor scenes; the Gray-ball dataset [38] of 11,340 images of diverse scenes, and the Cube+ dataset of 1365 images; which is a combination of the Cube dataset and an additional 342 images. The parameters are optimized through several experiments with Gehler and Shi datasets. The proposed RiR-SDN architecture is coded based on tensorflow [39] and implemented in NVIDIA TITAN RTX 24 G. The total learning time is 1.5 days with 10 K epochs. The network is designed to resize an image into 227×227 pixels and set an input batch to a batch size of 16. Figure 3 shows the convergence behavior of the proposed architecture. Here, basic training rates are compared in the experiment to find the optimal training rate for the proposed RiR-SDN architecture. Other parameters include a weight decay of $5\times 10^{5}$ and a momentum of 0.9. 

As highlighted above, the proposed approach has a primary difference from ResNet in that the proposed method uses the RiR-DSN architecture instead of a basic CNN block as in Figure 1. So, the proposed RiR-DSN architecture is deeper and more complex than the original residual block, resulting in a significant and meaningful improvement in estimation accuracy. Figure 4 shows the comparison results of the average angular error and the median angular error between the basic convolution and the proposed RiR-DSN architecture. In this comparative experiment, the basic convolution consists of ReLU [40], batch normalization [41], and a convolutional layer in order. The experiment uses the Gehler and Shi dataset, recording the median and average angular errors every 20 epochs. The proposed RiR-SDN architecture performs with much higher accuracy than basic convolution. Figure 4 also proves that estimation accuracy increases by removing batch normalization.

Figure 5 shows the resulting images from the proposed RiR-DSN architecture. All in (a) are the original images of the Gehler and Shi dataset. They are device-dependent raw images used as inputs for both training and cross-validation. Those images were taken with a Macbeth color chart placed in front for the purpose of evaluating the influence of the illuminant state on the image. All in (b) are the resulting images from estimating illumination with the proposed RiR-DSN architecture, and all in (c) are the ground truth images. Finally, all in (d) are the corrected images that are the goal of color constancy. It is worth noting that the greenish-blue illuminant colors are effectively removed from the original image (a), especially in the sky region. 

The following experiments compare the proposed RiR-SDN architecture versus its latest counterparts [42,43,44,45,46,47,48,49,50,51,52] with several standard datasets: Cube+, Gray-ball, and MultiCam datasets. The last several decades have seen extensive CVCC models proposed in the CVCC community. Despite significant advances in estimating illuminant color cast, there remain some limitations, such as algorithm failure or sudden accuracy drops under unusual circumstances. In neuroscience, it is a generally accepted fact that visual cortical neurons of human vision modulate receptive field sizes to adapt to changes in stimuli intensities, which is why human eyes perceive colors consistently regardless of changes in illumination conditions. However, this predominant trait of the HVS received little attention in the CVCC community, even though the primary focus of CVCC research has been on imitating how the HVS perceives constant colors of objects regardless of changes in illuminant conditions. Recently, learning-based approaches have been proposed to improve estimation accuracy. Choi et al. [53,54] put forward a novel approach that brings the residual network and atrous convolutions to the network structures. Choi and Yun [55] figure out another learning-based approach, named PMRN, that features the cascading mechanism and residual network within the DCNN architecture. However, a drawback of these approaches is that the filter sizes are fixed in the CNN, regardless of the intensity of the stimuli. Inspired by the color constancy ability of the HVS, the proposed RiR-DSN architecture uses SKCBs instead of a basic block, which consists of ReLU, batch-normalization, and a convolutional layer in order. The SKCBs used in the proposed architecture are able to scale up and down the filter kernel size, which is likened to the receptive field size modulated by the HVS automatically, to cope with changes in illumination sources. Table 1 is a summary of the comparative evaluation between various methods and the proposed architecture in terms of Mean, Median, Trimean, Best-25%, and Worst-25%. The results show the proposed RiR-DSN architecture outperforms its state-of-the-art counterparts.

Table 2 summarizes the experimental results that compare the proposed RiR-DSN architecture with its conventional counterparts. The results highlight a marked contrast between the proposed RiR-DSN architecture and its conventional counterparts. The proposed RiR-DSN causes the lowest angular error and is proved to be illuminant invariant. Table 3 summarizes other experimental results that evaluate inter-camera estimation accuracy with the MultiCam dataset of 1365 outdoor images captured by a Cannon 550D camera. The results demonstrate that the proposed RiR-DSN architecture surpasses its conventional counterparts in terms of inter-camera estimation accuracy, and also prove that the proposed RiR-DSN architecture is illuminant- and camera-invariant.

## 5. Conclusions

In achieving computer vision color constancy (CVCC), it is a vital but challenging problem to estimate unwanted scene illumination as accurately as possible. In an effort to improve accuracy as well as efficiency in estimating the illuminant color from a given image, this article presents a novel CVCC approach with RiR-DSN architecture. The DSN of the proposed RiR-DSN architecture is composed of selective kernel convolutional blocks (SKCBs), and the SKCBs, or neurons herein, are interconnected in a feed-forward fashion. In addition, the architecture has incorporated a dynamic selection mechanism into each SKCB, a neuron or layer of the convolutional neural network (CNN) herein, to ensure that the neuron can modulate filter kernel size depending on varying intensities of stimuli. In the CVCC space, it is a general observation that an increasing number of neurons and connections leads to performance improvement. Accordingly, the proposed RiR-DSN architecture is designed to be deeper and more complex compared to a basic convolutional layer, which consists of ReLU, batch normalization, and a convolutional layer in order. The comparative experiments use several standard datasets: Shi’s datasets, Cube + datasets, Gray-ball datasets, and MuliCam datasets, and the experiment results highlight that the proposed RiR-SDN architecture surpasses its conventional counterparts and makes unparalleled progress in both estimation accuracy and computational efficiency. In addition, the experimental results prove that the proposed RiR-SDN architecture is illuminant- and camera-invariant. Nevertheless, it is still crucial and worthwhile to continue to work towards optimizing convolutional neural networks and taking the CVCC to the next level.

## Figures and Tables

**Figure 1 sensors-23-05341-f001:**
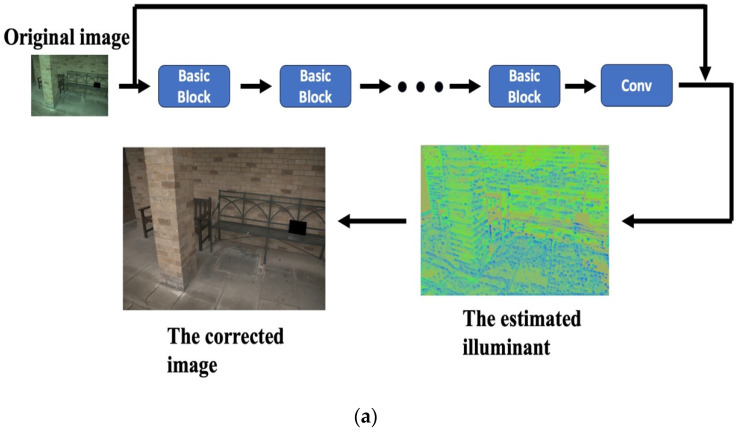
Typical (**a**) basic architecture and its (**b**) basic block.

**Figure 2 sensors-23-05341-f002:**
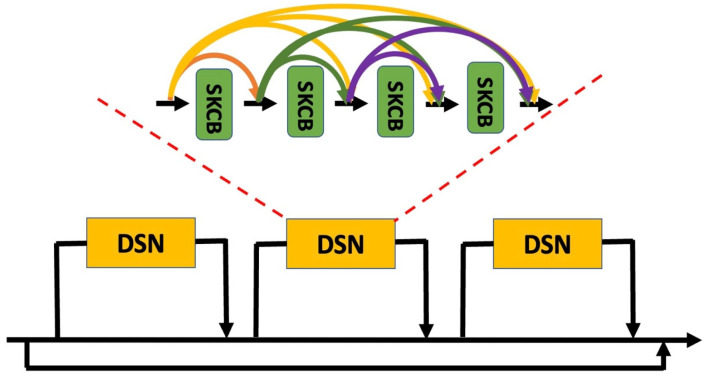
The proposed RiR-DSN architecture.

**Figure 3 sensors-23-05341-f003:**
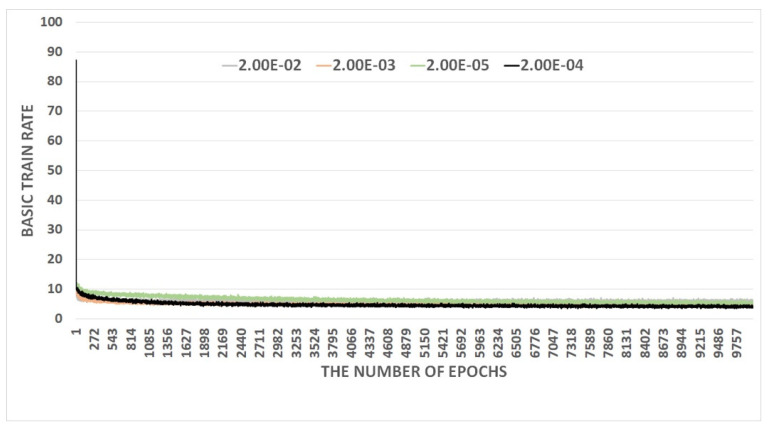
Comparison of basic training rates.

**Figure 4 sensors-23-05341-f004:**
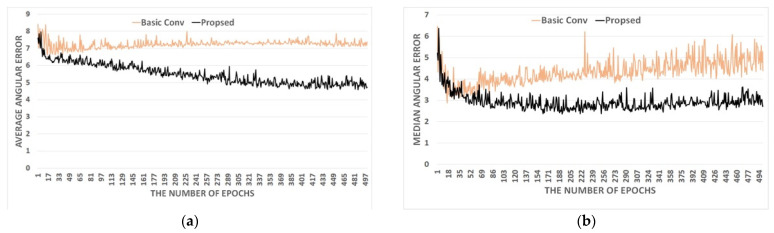
Accuracy comparison by (**a**) average angular error and (**b**) median angular error between the basic CNN and the proposed method.

**Figure 5 sensors-23-05341-f005:**
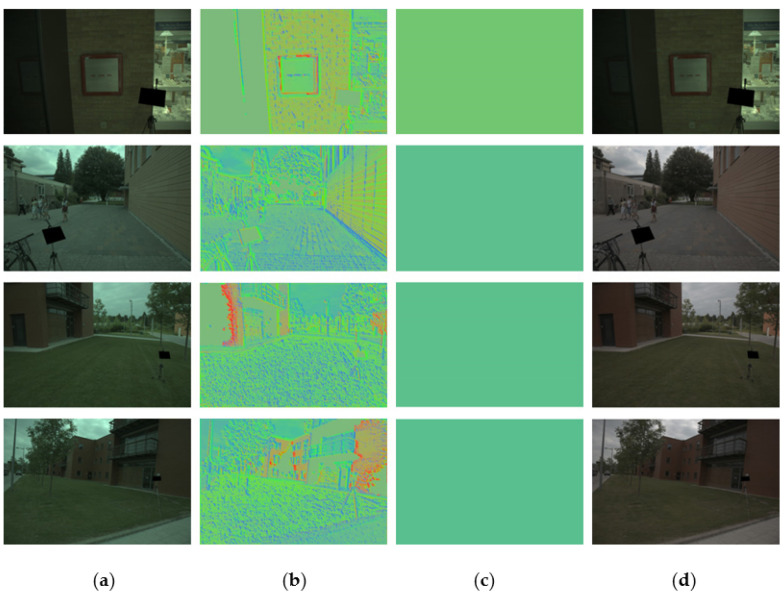
The resulting images from the proposed RiR-SDN architecture: (**a**) original image, (**b**) estimated illuminant, (**c**) ground truth image, and (**d**) corrected image with the Gehler and Shi dataset.

**Table 1 sensors-23-05341-t001:** Comparison of angular errors between various methods and the proposed architecture with Cube+ datasets (lower error means higher estimation accuracy).

Method (s)	Mean	Median	Trimean	Worst-25%	Best-25%
Statistics-Based Approach					
WP [5]	9.69	7.48	8.56	20.49	1.72
GW [4]	7.71	4.29	4.98	20.19	1.01
SoG [7]	2.59	1.73	1.93	6.19	0.46
First GE [8]	2.41	1.52	1.72	5.89	0.45
Second GE [8]	2.50	1.59	1.78	6.08	0.48
Learning-Based Approach					
FFCC [42]	1.38	0.74	0.89	3.67	0.19
FC4 (sque.) [43]	1.35	0.93	1.01	3.24	0.30
VGG-16 [44]	1.34	0.83	0.97	3.20	0.28
MDLCC [45]	1.24	0.83	0.92	2.91	0.26
One-net [46]	1.21	0.72	0.83	3.05	0.21
**Ours**	**1.13**	**0.60**	**0.80**	**2.55**	**0.18**

**Table 2 sensors-23-05341-t002:** Performance comparison of the proposed RiR-DSN architecture versus its counterparts with the Gray-ball dataset.

Method (s)	Mean	Median	Trimean	Best-25%	Worst-25%
**SVR [28]**	13.17	11.28	11.83	4.42	25.02
**BS [37]**	6.77	4.70	5.00	-	-
**NIS [17]**	5.24	3.00	4.35	1.21	11.15
**EM [47]**	4.42	3.48	3.77	1.01	9.36
**CNN [23]**	4.80	3.70	-	-	-
**Ours**	**2.87**	**1.59**	**1.66**	**0.47**	**5.98**

**Table 3 sensors-23-05341-t003:** Performance comparison of inter-camera illumination color estimation accuracy between the proposed RiR-DSN architecture and its counterparts with the MultiCam dataset.

Method (s)	Mean	Median	Trimean	Best-25%	Worst-25%
**GW [4]**	4.57	3.63	3.85	1.04	9.64
**PG [48]**	3.76	2.99	3.10	1.14	7.70
**WP [5]**	3.64	2.84	2.95	1.17	7.48
**1st GE [8]**	3.21	2.51	2.65	0.93	6.61
**2nd GE [8]**	3.12	2.42	2.54	0.86	6.55
**BS [37]**	3.04	2.28	2.40	0.67	6.69
**SoG [7]**	2.93	2.24	2.41	0.66	6.31
**SSS [49]**	2.92	2.08	2.17	0.46	6.50
**DGP [50]**	2.80	2.00	2.22	0.55	6.25
**QU [51]**	2.39	1.69	1.89	0.48	5.47
**CNN [23]**	1.88	1.47	1.54	0.38	4.90
**3-H [52]**	1.67	1.20	1.30	0.38	3.78
**FFCC [42]**	1.55	1.22	1.23	0.32	3.66
**FC4 (sque.) [43]**	1.54	1.13	1.20	0.32	3.59
**Ours**	**1.45**	**1.10**	**1.05**	**0.30**	**3.42**

## Data Availability

Available online: http://colorconstancy.com/.

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
