# Peer review of "CVCC Model: Learning-Based Computer Vision Color Constancy with RiR-DSN Architecture"

_sensors, 2023, doi:10.3390/s23115341_

Round 1
Reviewer 1 Report
The article presents a CVCC approach that introduces a residual-in-residual dense selective kernel network to achieve computer vision color constancy, and the experiment results show the effectiveness of improvement in this task. However, there are several aspects need to be explained and revised.
1. Fig.1, Fig.2. It is suggested to increase the content in the pictures to express the algorithm structure more clearly.
2. Fig.1. It seems like the detail of “Basic Block” in Fig.1 is not present in the article. It is suggested to add the description of it.
3. Section 3. It is suggested to add the detail of “selective kernel convolutional block (SKCB)” in the article.
Minor editing of English language required
Author Response
reviewer#1
- Fig.1, Fig.2. It is suggested to increase the content in the pictures to express the algorithm structure more clearly.
<Answer> Thanks for the suggestion. For better readability, the font size is increased and a phrase “Original image” is also added to indicate the input image.
- Fig.1. It seems likethe detail of “Basic Block” in Fig.1 is not present in the article. It is suggested to add the description of it.
<Answer> Thanks for the suggestion. The description of the basic block is added in the article.
- Section 3. It is suggested to add the detail of “selective kernel convolutional block (SKCB)” in the article.
<Answer> Thanks for the suggestion. The author detailed the “selective kernel convolutional block (SKCB)” in the article as follows:
“A SKCB is made up of three phases: Split, Fuse, and Select. The Split phase generates a multi-path for layers that use diverse filter kernel sizes. Next, at the Fuse phase, selection weights are given to the feature map information, and the weighted feature map information moves to the next layer. Finally, at the Select phase, all the feature map information merges [35].”

Reviewer 2 Report
This article presents a novel CVCC approach that introduces a residual-in-residual dense selective kernel network (RiR-DSN). It has a residual network in a residual network (RiR) and the RiR houses a dense selective kernel network (DSN). The architecture has incorporated dynamic selection mechanism into each neuron to ensure that the neuron can modulate filter kernel sizes depending on varying intensities of stimuli. Experimental results show that the RiR-DSN architecture performs well above its state-of-the-art counterparts.
The work is original. The contributions of the paper are clearly defined and detailed. It is a valuable study.
The manuscript is well-organized and well-written.
The followings can be fixed:
1- Providing a table that summarizes the related work would increase the understandability of the difference from the previous studies in the "Related Works" section.
2- A concern is that no formal statistical analysis of the results are done, to indicate whether the differences in performance are statistically significant or not.
For example; Friedman Aligned Rank Test, Wilcoxon Test, Quade Test, etc.
p-value can be calculated and compared with the significance level (p-value < 0.05).
3- Table 2 is explained with 2-3 sentences, similarly, Table 3 is explained with 2 sentences. They can be explained in more detail. The reasons of the results can also be discussed. For example; why it is the best? Why it is increasing or decreasing? etc.
4- The organization of the paper (the structure of the manuscript) may be written at the end of the "Introduction" section.
For example: "Section 2 presents ... Section 3 gives ...."
5- In the reference list, there are only four papers published after 2019.
I suggest the authors citing the most recent papers (especially published after 2019).
6- Some abbreviations are used in the text without giving their expansion.
For example; SoG, LPF, ECB, FFCC, FC4, VGG, MDLCC, SVR, NIS, DGP, etc.
The authors should write that "these abbreviations stand for what".
-
Author Response
Reviewer comment #2
1- Providing a table that summarizes the related work would increase the understandability of the difference from the previous studies in the "Related Works" section.
<Answer> The related work in the Previous Works section is too big to fit in a summary table.
2- A concern is that no formal statistical analysis of the results are done, to indicate whether the differences in performance are statistically significant or not.
For example; Friedman Aligned Rank Test, Wilcoxon Test, Quade Test, etc.
p-value can be calculated and compared with the significance level (p-value < 0.05).
<Answer> The statistical significance of experimental results is evaluated by mean, trimean, median, best 25% and worst 25% of the angular errors, which are the formal statistical analysis in the color constancy space.
3- Table 2 is explained with 2-3 sentences, similarly, Table 3 is explained with 2 sentences. They can be explained in more detail. The reasons of the results can also be discussed. For example; why it is the best? Why it is increasing or decreasing? etc.
<Answer> Table 1 elaborates the experimental results and their discussion. Table 2 and 3 are additional experimental results, which are designed to prove that the proposed method is camera invariant and illuminant invariant.
4- The organization of the paper (the structure of the manuscript) may be written at the end of the "Introduction" section.
For example: "Section 2 presents ... Section 3 gives ...."
<Answer> Thanks for the suggestion. The organization of the paper is written at the end of the “Introduction” section as follows:
“The Section 2 discusses the previous works in detail and the Section 3 elaborates the proposed RiR-DSN architecture for achieving color constancy. The Section 4 evaluates the quantitative and qualitative results of estimating the illuminant with the proposed architecture. Finally, the Section 5 is the conclusions of this paper.”
5- In the reference list, there are only four papers published after 2019. I suggest the authors citing the most recent papers (especially published after 2019).
<Answer> Thanks for the suggestion. The author added the most recent papers in the reference list, especially published after 2019 as follows:
- H. H. Choi; H. S. Kang; B. J. Yun, CNN-based illumination estimation with semantic information. Appl. Sci., vol. 10, no. 14, pp. 1-17, Jul. 2020.
- H.-H. Choi; B.-J. Yun, Deep learning-based computational color constancy with convoluted mixture of deep experts (CMoDE) fusion technique. IEEE Access, vol. 8, pp. 188309-188320, 2020.
- H.-H. Choi; B.-J. Yun, Learning-based illuminant estimation model with a persistent memory residual network (PMRN) architecture. IEEE Access, vol. 9, pp. 29960-29969, 2021.
6- Some abbreviations are used in the text without giving their expansion.
For example; SoG, LPF, ECB, FFCC, FC4, VGG, MDLCC, SVR, NIS, DGP, etc.
The authors should write that "these abbreviations stand for what".
<Answer> In response, the author provided the expansion of each abbreviation.

Reviewer 3 Report
In this article, the author presented the usefulness of RiR-DSN architecture for the CVCC model in AI images. It offers a low angular error and is proven to be illuminant invariant. It is also composed of a selective kernel convolutional block. These enhance the conventional CVCC model for the next generation. It is a well-written article. There are no significant comments. I propose that the author make a flowchart of the history of CVCC architecture to understand it well to general readers.
Moderate English editing is required.
Author Response
Reviewer #3
In this article, the author presented the usefulness of RiR-DSN architecture for the CVCC model in AI images. It offers a low angular error and is proven to be illuminant invariant. It is also composed of a selective kernel convolutional block. These enhance the conventional CVCC model for the next generation. It is a well-written article. There are no significant comments. I propose that the author make a flowchart of the history of CVCC architecture to understand it well to general readers.
<Answer> Many thanks for the reviewer’s kind comment. Regarding the reviewer’s proposal to make a flowchart of the history of CVCC architecture, the Section 2 provides the history of CVCC architecture, discussing the five CVCC categories in chronological order.

Round 2
Reviewer 1 Report
I think this paper can be accpeted.
None